# VDAC1 at the Intersection of Cell Metabolism, Apoptosis, and Diseases

**DOI:** 10.3390/biom10111485

**Published:** 2020-10-26

**Authors:** Varda Shoshan-Barmatz, Anna Shteinfer-Kuzmine, Ankit Verma

**Affiliations:** Department of Life Sciences and the National Institute for Biotechnology in the Negev, Ben-Gurion University of the Negev, Beer-Sheva 84105, Israel; shteinfe@post.bgu.ac.il (A.S.-K.); verma@post.bgu.ac.il (A.V.)

**Keywords:** apoptosis, cancer, diseases, metabolism, mitochondria, VDAC1, virus

## Abstract

The voltage-dependent anion channel 1 (VDAC1) protein, is an important regulator of mitochondrial function, and serves as a mitochondrial gatekeeper, with responsibility for cellular fate. In addition to control over energy sources and metabolism, the protein also regulates epigenomic elements and apoptosis via mediating the release of apoptotic proteins from the mitochondria. Apoptotic and pathological conditions, as well as certain viruses, induce cell death by inducing VDAC1 overexpression leading to oligomerization, and the formation of a large channel within the VDAC1 homo-oligomer. This then permits the release of pro-apoptotic proteins from the mitochondria and subsequent apoptosis. Mitochondrial DNA can also be released through this channel, which triggers type-Ι interferon responses. VDAC1 also participates in endoplasmic reticulum (ER)-mitochondria cross-talk, and in the regulation of autophagy, and inflammation. Its location in the outer mitochondrial membrane, makes VDAC1 ideally placed to interact with over 100 proteins, and to orchestrate the interaction of mitochondrial and cellular activities through a number of signaling pathways. Here, we provide insights into the multiple functions of VDAC1 and describe its involvement in several diseases, which demonstrate the potential of this protein as a druggable target in a wide variety of pathologies, including cancer.

## 1. Mitochondria as Signaling Hubs

Mitochondria as cellular energy powerhouses provide a central location for the multiple metabolic reactions required to satisfy the energy and biomolecule demands of cells, and serve to integrate the diverse metabolic pathways and provide cells with metabolic flexibility. The appreciation of mitochondria as sites of biosynthesis and bioenergy production has dramatically expanded in recent years, and they are now known to play a crucial role in almost all aspects of cell biology and to regulate cellular homeostasis, metabolism, innate immunity, apoptosis, epigenetics, cellular fate, and more [1,2]. Mitochondrial dysfunction induces stress responses, which link the fitness of this organelle to the condition of the whole organism. These functions typically center around the metabolic traffic in and out of the mitochondria, and they are likely to contribute to the exceptional variability of mitochondrial disease manifestations. The voltage-dependent anion channel 1 (VDAC1) protein, which is located in the outer mitochondrial membrane (OMM), serves as a gatekeeper that can regulate the metabolic and energetic cross-talk between mitochondria and the rest of the cell, and plays a role in mitochondria-mediated apoptosis [3,4,5].

Other different aspects of VDAC1 structures [6] and functions, such as in cell stress [5], Ca^2+^ regulation [7,8], metabolism [5], and apoptosis [7,9], and as a therapeutic target [10,11,12,13,14,15], were presented in ours and others’ recent reviews. 

## 2. VDAC1 Structural Elements: The N-Terminal Domain and Its Oligomeric State

Three mammalian isoforms of VDAC (VDAC1, VDAC2, and VDAC3) that share a number of functional and structural attributes have been identified [16,17,18]. Information about VDAC isoform function and structure was obtained from channel activity of purified and reconstituted protein and, using cell-based assays for survival, metabolism, reactive oxygen species (ROS), and cellular Ca^2+^ regulation, and by gene knockout mouse models [17]. VDAC1 is the most abundant isoform and the focus of this review. VDAC2 knock out is lethal and is considered to be an anti-apoptotic protein. VDAC3 is the least known, active as channel. While VDAC 1 contains two cysteines, VDAC2 and VDAC3, with nine and six cysteines, respectively, are proposed to function as oxidative stress sensors [17]. The crystalline structure of the most prevalent and studied isoform, VDAC1, was solved at atomic resolution, revealing a β-barrel composed of 19 transmembrane β-strands connected by flexible loops. The β1 and β19 strands, together, are arranged in parallel, where the N-terminal region (26-residues) lies inside the pore [19,20,21], but can flick out of it [22,23] and interact with hexokinase (HK) [4,5,24,25,26,27,28,29,30], Aβ [31,32], and other proteins, such as the anti-apoptotic proteins, Bcl-2 and Bcl-xL [4,27,33,34,35,36,37]. Thus, this region of the protein is well positioned to regulate the traffic of materials through the VDAC1 channel [19,21].

The diameter of the channel pore has been estimated as 2.6–3.0 nm [19], but this can decrease to about 1.5 nm when the N-terminal flexible region is located inside the pore [19,20,21]. The sequence of rich with glycine residues (21GlyTyrGlyPheGly25) [19,20,21] connecting the N-terminal region to β1 strand is thought to confer the flexibility needed for this region to move in and out of the VDAC1 channel [23]. This mobility has been reported to be important for channel gating, dimerization of VDAC1 [23], and interaction with HK and members of the apoptosis regulating Bcl-2 family (i.e., Bax, Bcl-2, and Bcl-xL) [23,27,33,34,38,39].

Membranal and purified VDAC1, can form dimers, trimers, tetramers, hexamers, and higher-order oligomeric forms [4,30,40,41,42,43,44,45,46,47] through contact sites that have been identified [48]. We have demonstrated this oligomerization to be a dynamic process that occurs in response to a variety of apoptotic stimuli, acting through a range of signaling processes [28,40,42,44,45,47,49,50,51,52,53,54], as presented below (Section 5.1).

## 3. VDAC1 Extra-Mitochondrial Localization, Function, and Association with Pathological Conditions 

In addition to the mitochondrial membrane, VDAC has also been detected in other cell compartments [16,55,56,57,58,59,60,61,62,63,64,65], including the plasma membrane [16,57], the sarcoplasmic reticulum (SR) of skeletal muscles [66], and the endoplasmic reticulum (ER) of the rat cerebellum [64,67].

Antibodies raised against the N-terminus of VDAC1 interacted with the plasma membrane of bovine astrocytes and blocked a high conductance anion channel [61]. Interestingly, when detected in the plasma membrane (pl-VDAC1), the amino acid residues that were exposed to the cytosol in the mitochondrial protein were found to face the extracellular space [55,60,68,69,70,71]. This was demonstrated in epithelial cells, astrocytes, and neurons [56,57] and in differentiated hippocampal neurons [55]. VDAC1 has also been identified in the brain post-synaptic membrane fraction [60] and in the caveolae or caveolae-related domains of established T lymphoid-like cell lines [58]. The protein has also been found to participate in the multi-protein complexes found in lipid rafts, together with the estrogen receptor α (mERα) and insulin-growth factor-1 receptor (IGF-1R) [58,62,63,64]. In red blood cells that do not possess mitochondria, VDAC1 has been found in the endoplasmic reticulum, and Golgi apparatus [59].

The plasma membrane form of VDAC1 may have an extended N-terminal signal peptide which is responsible for its targeting to the cell membrane [71,72]. Alternatively, the human plasminogen kringle 5 (K5) may induce translocation of VDAC1 to the cell surface, where the protein was recently identified as the receptor for K5 on HUVEC membrane [73,74]. Additional mechanisms, such as the presence of alternative mRNA untranslated regions, have also been suggested [57,64].

The levels of pl-VDAC1 were reported to be increased under pathological conditions such as Alzheimer’s disease (AD) [75,76,77], where it has been suggested that pl-VDAC1 serves as an “amyloid-regulated” apoptosis related channel [62,63,65]. We recently described a direct interaction between Aβ and the N-terminal region of VDAC1, and demonstrated that VDAC1 is required for Aβ entry into the cells [78,79] and apoptosis induction [31,32]. We have also recently demonstrated that VDAC is overexpressed in type 2 diabetes (T2D) [80,81,82,83] and mistargeted to the β-cell plasma membrane [84]. This overexpression under pathological conditions is also seen in cancer [3,15,85,86], autoimmune diseases such as lupus [87], non-alcoholic steatohepatitis (NASH) [88], inflammatory bowel disease (IBD) (unpublished data), and cardiac diseases [89,90], as presented below (Section 8).

The exact functions of extra-mitochondrial VDAC are unknown, although several possible roles have been proposed (reviewed in [64,91,92]), and include regulation of tissue volume in the brain [61], and other cell types [70,93,94], or release of ATP in β-cells [84] and in human erythrocytes [95].

## 4. VDAC1, a Multi-Functional Channel Controlling Cell Energy, Metabolism, and Oxidative Stress

To reach the mitochondrion matrix or to be released to the cytosol, all metabolites and ions must traverse the OMM via VDAC1, the sole channel mediating the flux of ions, nucleotides, and other metabolites up to ~5000 Da. In this way, VDAC maintains control of the metabolic and ion cross-talk between the mitochondria and the rest of the cell (Figure 1). Nucleotides and metabolites transported include pyruvate, malate, succinate, and NADH/NAD+, as well as lipids, heme, cholesterol, and ions such as Ca^2+^ [3,4,5,96]. In contrast, there are over 50 mitochondrial substrate-specific carrier proteins of the family solute carrier family 25 (SLC25) in the inner mitochondrial membrane (IMM), such as the (ADP/ATP) antiporter, the adenine nucleotide translocator (ANT), the transporter of Pi (PiC), as well as transporters of aspartate/glutamate, pyruvate, acyl carnitine, and citrate, among others [97].

Closure [98] or down-regulation of VDAC1 channel expression reduced the exchange of metabolites between the mitochondria and the rest of the cell and inhibited cell growth [86,99], indicating the importance of the protein to the maintenance of physiological cellular function.

As already described, the mitochondria are the energy source of the cell. They are responsible for the ATP generated during glycolysis and oxidative phosphorylation (OXPHOS). This is then exported to the cytosol and exchanged for ADP, which is recycled again in the mitochondria to generate ATP. This shuttling process that also involves ANT and creatinine kinase (CrK), located between the IMM and OMM [100], and may be regulated by tubulin αβ heterodimers [101], but it is ultimately facilitated by VDAC1, which thereby controls the electron transport chain [4] (Figure 1) and the energy state of the cell [98].

In addition to ATP, VDAC1 is also involved in the transfer of a number of other essential molecules across the OMM. These include Ca^2+^, cholesterol, fatty acids, and reactive oxygen species (ROS). Controlling the flow of Ca^2+^ allows VDAC1 to regulate mitochondrial Ca^2+^ homeostasis, oxidative phosphorylation, and Ca^2+^ cross-talk between the VDAC1 in the OMM and the IP3 receptor in the ER. This takes place through the mitochondria associated membranes (MAM) and involves the chaperone GRP75 [67,102,103].

VDAC1 [104] is a necessary component of the cholesterol transport multi-protein complex, the transduceosome, composed of the translocator protein (TSPO), and the steroidogenic acute regulatory protein (STAR) [88,105] (Figure 1).

VDAC1 is also part of a complex mediating the transport of fatty acids through the OMM [88,105,106] composed of carnitine palmitoyltransferase 1a (CPT1a), which faces the intermembrane space (IMS), and the long chain acetyl coenzyme-A (acyl-CoA) synthetase (ACSL) protein. Once activated by ACSL, VDAC1 transfers acyl-CoAs across the OMM to the IMS, where they are converted into acylcarnitines by CPT1a (Figure 1). They are then transferred across the IMM by carnitine/acylcarnitine translocase, and converted back into acyl-CoA by CPT2 in the IMM, and, subsequently, undergo β-oxidation in the matrix [105,107]. In this respect, VDAC1 has recently been reported to serve as a lipid sensor [108]. Finally, VDAC is also involved in regulating oxidative stress [5]. ROS formed by reaction with O_2_•^−^ at complex III are released through VDAC1 where they activates c-Jun N-terminal kinase (JNK), the extracellular signal-regulated kinase (ERK 1/2), and p38, members of the mitogen-activated protein kinase (MAPK) family of serine/threonine kinases whose signaling may be detrimental to mitochondrial function [109,110]. Importantly, ROS release and consequent cytotoxicity are decreased when HK-I and HK-II bind to VDAC1 [111,112,113,114].

VDAC1 is also affected by hypoxic conditions. The C-terminal end of VDAC1 is cleaved (VDAC1-ΔC), with silencing of hypoxia inducible factor 1A (HIF-1α) prevents such cleavage [115,116]. This formation of VDAC1-ΔC, is thought to prevent apoptosis and permit the maintenance of ATP and cell survival in hypoxia [117].

As described, the location of VDAC1 in the OMM provides the perfect opportunity to preside over the traffic of metabolites between the mitochondria and the cytosol, where it interacts with other proteins in order to orchestrate and integrate mitochondrial functions with other cellular activities [3,4,30,41,42,118,119] (Figure 1).

VDAC1 activities are modulated by Ca^2+^, ATP, glutamate, and NADH, as well as by a variety of proteins (see Section 7) [120,121,122,123]. Using a photo-reactive ATP analog, we identified three potential nucleotide binding sites [122]. Subsequent NMR spectroscopy and site-directed mutagenesis revealed that hVDAC1 possesses one major binding region for ATP, UTP, and GTP that is formed by the N-terminal α-helix, the linker connecting the helix to the first β-strand, and adjacent barrel residues [124]. The crystal structure of mouse VDAC1 in the presence of ATP, revealed an additional low-affinity binding site [125]. With respect to a high and low affinity ATP binding site, it should be noted that the cellular concentration of ATP is 1–2 mM.

In addition, Ca^2+^ binds to VDAC1 although the physiological function of this connection is not clear. Binding of Ca^2+^ to purified and bilayer reconstituted VDAC1 maintained the channel in an open configuration, which could be useful in upregulating the exchange of metabolites [126]. The divalent cation-binding sites bind the lanthanides, La^3+^ and Tb^3+^, as well as ruthenium red (RuR), and its analogue Ru360 [127,128,129], the photo-reactive analogue azido ruthenium (AzRu) [130]. All reduce the conductance of native, but not mutant, VDAC1.

VDAC1 undergoes all known types of post-translational modifications (PTMs), including nitrosylation, acetylation, carbonylation, and phosphorylation (124). TVDAC1 contains two cysteines; Cys^232^ is found in the carboxyamidomethylated form, while Cys^127^ is in the oxidized form of sulfonic acid [131]. VDAC1 possesses several potentially phosphorylatable serine and threonine residues, many of which have indeed been shown to undergo phosphorylation [132,133,134] by protein kinase A (PKA) [133], protein kinase C (PKC)ε [132], and GSK3b [135]. Both VDAC1 and VDAC2 are phosphorylated at a specific Tyr residue under hypoxic conditions [134].

Under pathological conditions, such as oxidation, aging, or after ischemic reperfusion injury, VDAC was shown to undergo nitration [136,137,138], while the protein undergoes carbonylation in the Alzheimer’s disease-affected brain or after exposure to acrolein, produced by lipid peroxidation [139].

## 5. VDAC1, at the Nexus of Mitochondria-Mediated Apoptosis, and mtDNA Release 

Mitochondria play a crucial role in the induction of apoptosis [140]. Changes in the permeability of the mitochondrial membrane in response to an apoptotic signal facilitate the release of apoptogenic proteins, including apoptosis-inducing factor (AIF), cytochrome *c* (Cyto *c*), and second mitochondria-derived activator of caspase and Direct lnhibitor of apoptosis-cinding protein with low pI (SMAC/Diablo) crossing the OMM into the cytoplasm [3,140]. Once in the cytosol, Cyto *c* and SMAC/Diablo stimulate a caspase activating cascade, and AIF translocates to the nuclease where it activates nucleases and initiates a sequence of events leading to the degradation of proteins and DNA, and cell death. The release of the apoptogenic proteins is thought to be through a channel in the mitochondrial membrane, which may be composed of Bax and/or Bak oligomers [141,142], hetero-oligomers of VDAC1 and Bax [143,144], or VDAC1 oligomers [28,30,33,40,41,42,47,50,51] (Figure 1) (for reviews, see [4,28,30]). In this context, reconstituted liposomes containing purified VDAC1 have been shown to release encapsulated Cyto *c* to the external medium [37,47,145].

VDAC1 promotes apoptosis by mediating the release of apoptosis-causing proteins through an oligomeric channel. However, VDAC1, can also regulate the process by interacting with anti-apoptotic proteins [3,4,5,42] providing an additional measure of apoptosis control.

The crucial role of VDAC1 in apoptosis induction is demonstrated by the effects of VDAC1 silencing or over-expression [4,24,29,33,44,146,147,148]. Over-expression of VDAC1 was shown to induce apoptosis in all cell types [24,29,51,146,147,148], and cell death could be prevented by reagents such as RuR [29,149], 4,4 diisothiocyanostilbene-2,2-disulfonic acid (DIDS), 4-acetamido-4-isothiocyanato-stilbene-2,2-disulfonic acid (SITS), 4,4′ diisothiocyanatodihy-drostilbene-2,2′-disulfonic acid (H_2_DIDS), 4,4′-dinitrostilbene-2,2′-disulfonic acid (DNDS), or diphenylamine-2-carboxylate (DPC) [54] that interact with VDAC1 [24,29,33,44,51,146,147,148], or by the overexpression of anti-apoptotic proteins such as Bcl2 and HK-I [24,26,29]. As a corollary, reducing VDAC1 expression with siRNA negated cisplatin-induced apoptosis and Bax activation in non-small cell lung cancer (NSCLC) cells [150], reduced apoptosis induced by endostatin [151], and inhibited selenite-induced permeability transition pore (PTP) opening in HeLa cells [152].

### 5.1. Oligomerized VDAC1 Releases Cyto c, AIF, and SMAC/Diablo

Nucleotides and small molecules up to 5 kDa can pass through the VDAC1 pore, but not a folded protein like Cyto *c* (12 kDa). Therefore, we proposed that VDAC1 oligomerization can form a large channel, able to export pro-apoptotic proteins IMS to the cytoplasm to initiate apoptosis [24,28,30,33,40,41,42,44,47,50,51,145]. This change in VDAC1’s structural state occurs in response to pro-apoptotic stimuli such as curcumin, As_2_O_3_, etoposide, cisplatin, selenite, H_2_O_2_, or UV light. Even in the absence of stimuli, overexpression of VDAC1 shifts the equilibrium towards oligomerization and, this led to apoptosis in several tested cell types [24,29,33,44,51,146,147,148].

The oligomerization process is significantly affected by the lipid composition of the OMM [153], and is enhanced by p53 [154].

As already described, a number of materials that interact with VDAC1 can interfere with oligomerization and attenuate the resultant apoptosis. Two new compounds we have developed, AKOS-022 and VBIT-4, similarly interact with VDAC1 to reduce its oligomerization and subsequent apoptosis induced by a variety of stimuli in various cell lines [53]. Importantly these new molecules protect against mitochondrial dysfunction, specifically restoring the membrane potential of the mitochondria, which is altered by apoptotic stimuli, thereby reversing energy and metabolic changes, decreasing the production of ROS, and maintaining physiological levels of intracellular Ca^2+^. Inhibiting the initiation of apoptosis at an early stage by inhibiting VDAC1 oligomerization may represent an effective approach to prevent or slow the enhanced apoptosis seen in neurodegenerative disorders [155,156] and various cardiovascular diseases [157,158,159].

All of these observations suggest the presence of a dynamic equilibrium in which VDAC1 shifts from a monomeric form towards oligomerization in response to pro-apoptotic stimuli or VDAC1 overexpression. In this way, the levels of VDAC1 expression and oligomerization are possible targets for intervention in mitochondrial-mediated apoptosis.

### 5.2. Release of Mitochondrial DNA (mtDNA) through the VDAC1 Channel

Human mitochondrial DNA (mtDNA) is a circular molecule of 16,569 bp, that encodes 13 polypeptides, as well as the 22 tRNAs and two rRNAs that are required for their translation [160]. The noncoding region (NCR) regulates transcription and translation of the mtDNA.

Many mtDNA genomes possess a third, linear strand region in the major NCR, which forms a triple-stranded structure known as the mitochondrial displacement loop, or D-loop. Although the precise function of the mitochondrial D-loop is unclear, a number of proposed options include DNA topology, replication, recombination, membrane association and dNTP metabolism [161]. The region is known to include promoter for adjacent transcripts possessing a high level of sequence variability, and has been associated with certain cancers, as well as aging in skeletal muscles, and skin fibroblasts [161].

Alteration in mtDNA content can impair cell metabolism and the innate immune system, including elevating the expression of interferon-stimulated genes [162].

Recently, we demonstrated that short mtDNA fragments, corresponding to a region within the D-loop, crossed the OMM and were released into the cytosol via the oligomerized VDAC1 channel and then triggered type-Ι interferon signaling and increased disease severity in a systemic lupus erythematosus (SLE) mouse model [87]. We also showed that our novel molecule VBIT-4 could inhibit this process [87]. Thus, inhibiting VDAC1 oligomerization and preventing mtDNA from passing through the OMM may represent a novel strategy for treating not only autoimmune diseases, such as SLE, but also other diseases.

A number of studies have now reported that viruses damage the host cell mtDNA to control the cell. For example, the Zta protein encoded by the Ebola virus translocates into the mitochondria and affects mtDNA replication [163]. Similarly, hepatitis C virus (HCV) infection leads to mtDNA damage [164], and the herpes simplex virus produces UL12.5 protein leading to mtDNA degradation [165], while HIV and hepatitis C virus infections cause mtDNA depletion in co-infected patients [166]. 

These observations are interpreted to indicate the importance of the mtDNA and the integrity of mitochondria and their metabolism for a variety of human pathologies.

### 5.3. VDAC1 Overexpression and Induction of Apoptosis 

As already described, overexpression of VDAC1 is associated with apoptosis [3]. Levels of VDAC1 expression may be increased by a variety of agents, for example, in A375 human malignant melanoma cells by the tyrosinase inhibitor arbutin (hydroquinone-O-beta-D-glucopyranoside) [167], but increases in expression were also seen in acute lymphoblastic leukemia (ALL) cell lines following prednisolone treatment [168]. Other examples are listed in Table 1. Interestingly, cisplatin upregulated VDAC1 expression in a cisplatin-sensitive cervix squamous cell carcinoma cell line (A431), but down-regulated VDAC1 in a cisplatin-resistant cell line (A431/Pt) [169]. Pyrroloquinoline quinone (PQQ), an essential micronutrient [170] found in fruits and vegetables and cocoa powder has also been demonstrated to increase VDAC1 expression, but this was attributed to the induction of mitochondrial biogenesis [171].

VDAC1 has been identified as a receptor of plasminogen kringle 5 (K5) known to display a potent anti-angiogenesis effect through inducing endothelial cell (EC) apoptosis. It is proposed that K5 induces apoptosis by up-regulating VDAC1 expression level via inhibiting the ubiquitin-dependent degradation of VDAC1 by promoting its phosphorylation through AKT-GSK3 pathway. In addition, K5 promoted translocation of VDAC1 to the plasma membrane where it binds to K5 [74].

The mechanism for up-regulation of the expression of VDAC1 is thought to involve an increase in the intracellular concentration of ionized calcium ([Ca^2+^]i), [7,8,50,51]. Increasing the [Ca^2+^]i directly by the addition of A23187, ionomycin, or thapsigargin also upregulated VDAC1 expression, and promoted oligomerization, and apoptosis [50,51,109]. As a corollary, chelating Ca^2+^ with BAPTA-AM and thereby reducing [Ca^2+^]i, inhibited the overexpression and oligomerization of VDAC1, and subsequent apoptosis. This process of overexpression of VDAC1 induced by apoptosis stimuli, stress conditions, and pathological conditions leading to oligomerization and triggering apoptosis [40,41] is presented in Figure 2.

The increase in VDAC1 transcription is a result of modulation of the interaction between transcription factors (TFs) and the VDAC1 promoter, or possibly by alterations in the expression of specific microRNA. We therefore propose that the expression of VDAC1 may represent a new possibly general mechanism of action by which inducers of apoptosis and pathological conditions upregulate the expression of VDAC1 (Figure 3).

## 6. VDAC1, Metabolism, and a Link to Epigenetics

Epigenetic changes involving methylation of DNA and/or modification to histones by acetylation, ubiquitination, methylation, phosphorylation, SUMOylation, glycosylation, or biotinylation represent an innate mechanism that links nutritional status to gene expression, and play a critical role in many cellular processes [180,181,182,183,184,185,186]. The enzymes responsible for these modifications include histone acetyltransferases (HATs), histone deacetylases (HDACs), methyltransferases (KMTs), and demethylases (KDMs) [187,188,189].

Most chromatin-modifying enzymes utilize metabolites as co-factors or substrates, and, thus, are directly dependent on metabolites such as acetyl-CoA, citrate, ketoglutarate, and NAD+, among others. Therefore, by supplying these metabolites, mitochondria can regulate the expression of different genes to facilitate diverse cellular functions [190]. Metabolites of the TCA cycle are also involved in controlling chromatin modifications, DNA methylation, hypoxic response, immunity, and post-translational modifications of proteins, and can thereby influence the function [191]. Since such metabolites require VDAC1 for their transport to the cytosol in order to reach the nucleus, VDAC1 is thus a critical regulator of gene expression [192] (Figure 4).

Relevant metabolites include: Acetyl-CoA is a required co-factor for enzymes such as histone acetyltransferases (HATs) that catalyze the transfer of an acetyl group, to form ɛ-N-acetyl-lysine and acetylate histones in a process that is known to alter chromatin dynamics and to drive the epigenetic control of gene expression by activating transcriptional programs [193]. Practically, histone acetylation is dynamically regulated by the opposing actions of HATs and histone deacetylases (HDACs) that catalyze the addition and removal of the acetyl group, respectively. Intracellular concentrations of acetyl-CoA can vary roughly ~10-fold under normal physiological conditions, but they fall within the Km range of HATs. Histone acetylation activity is, thus, essentially regulated by the availability of acetyl CoA.NAD+ is essential for the deacetylation activity of sirtuins, a subgroup of HDAC, and changes in the NAD+/NADH ratio are thought to positively regulate sirtuin activity.Citrate in the cytosol can be converted to acetyl-CoA by the enzyme ATP-citrate lyase (ACLY), which catalyzes the ATP-dependent cleavage of mitochondrial-derived citrate into oxaloacetate and acetyl-CoA [194]. The acetyl-CoA can then serve as a donor for HAT-mediated histone acetylation. However, citrate can also be converted to acetyl-CoA in the nucleus by ACLY [194].α-ketoglutarate (α-KG) is an essential co-factor for 2-oxoglutarate-dependent dioxygenases (2-OGDD), including the histone demethylases with a Jumonji domain (JMJDs) and ten-eleven translocation (TET) DNA demethylases. α-KG has a direct impact on gene expression and, thus, can influence cellular fate by regulating histones and DNA demethylases. Succinate is the product of 2-oxoglutarate dependent dioxygenase (2-OGDD) enzyme reactions and, thus, when it accumulates, it works as an antagonist to the reaction.The 2-Hydroxyglutarate (2-HG) is not part of the TCA cycle, but it can be derived from α-KG by enzymes in the mitochondrial matrix and cytosol. The 2-HG competitively inhibits 2-OGDDs and together with fumarate, can rewire the epigenetic landscape of the cells through inhibition of histone and DNA demethylases.

The 2-HG inhibits the activity of α-KG dependent dioxygenases such as TETs and JMJDs, which has broad implications for the regulation of epigenome.

6.Succinate and fumarate both behave as α-KG competitive antagonists and inhibit TETs and JMJDs. Both of these metabolites inhibit TET-catalyzed hydroxylation of 5 mC and the activity of histone demethylases KDM2A and KDM4A [195].7.S-adenosylmethionine (SAM) serves as the methyl donor in reactions catalyzed by methionine adenosyl-transferase (MAT). DNA methylation at CpG sites represses gene expression by impeding access to transcription factors and inhibition of RNA polymerase II.

Methylation markers on lysine residues in histone proteins also have a key role in regulating chromatin structure and gene transcription. Multiple lysine residues (H3K4, H3K9, H3K27, H3K36, H3K79, etc.) may be mono-, di- or tri-methylated, giving rise to a very complex histone methylation profile.

The dynamics of DNA and histone methylation are also regulated by the activity of DNA and histone demethylases, respectively. Demethylation of histone lysine marks are mediated by flavin-dependent histone lysine demethylases that consist of lysine-specific protein demethylases (KDM1) and JMJD enzymes, and the ten-eleven translocation hydroxylases (TET1-3).

We recently [192] demonstrated that depleting VDAC1 in glioblastoma cells U87-MG-derived tumors affects the metabolism–epigenetics axis of the tumor. Analysis with DNA microarrays, q-PCR, and specific antibodies revealed epigenetic alterations in the methylation and acetylation of histones and the levels of epigenetic-related enzyme levels following depletion of VDAC1 in these cells. These findings support the importance of VDAC1, as a transporter of epigenetic associated metabolites. Depletion of VDAC1 down-regulates mitochondrial metabolism [43,51,99,196] because of the decrease in substrates transported into the mitochondria and the inability of the produced metabolites to exit the mitochondria. This restricts the availability of substrates for chromatin modifications and affects the interplay between metabolism and epigenetics.

## 7. Proteins Interacting with VDAC1 Modulate VDAC1 Activity and They Are Regulated by VDAC1

As already discussed, VDAC1 may be considered a hub protein that interacts with more than 100 proteins associated, on one hand, with cell survival and, on the other, with apoptotic cell death [4,30,41,42]. Together, they integrate mitochondrial activities with other cellular processes [197]. The interacting proteins may be located in the OMM, IMM, IMS, cytosol, ER, plasma membrane, or/and nucleus [60,66]. Thus, VDAC1 in the OMM is a convergence locus for signals concerning the fate of the cell [4,41,42,43,198].

Importantly, we have been able to develop VDAC1-based peptides, which can interfere with these interactions, leading to impaired cell metabolism and apoptosis [27,34,39,199,200].

### 7.1. VDAC1 Interacting Metabolism-Related Proteins

VDAC1 interacts with a large number of metabolism-related proteins. Mitochondrial-bound HK (HK-I and HK-II) is overexpressed in cancer [4,201,202]. As a rate-limiting enzyme of glycolysis, HK association with VDAC1 offers several advantages to cancer cells [4,43]. HK binding to VDAC1 [24,25,26,27,28,29] allows direct coupling of mitochondrially generated ATP to glucose phosphorylation. Thus, the formation of a VDAC1-HK complex coordinates glycolytic flux with the actions of the TCA cycle and ATP synthase [4,44,201].

The formation of the HK-VDAC1 complex is regulated by Akt [203] and glycogen synthase kinase 3 beta (GSK3β), while the HK-VDAC complex is disrupted by VDAC phosphorylation [204]. The physical interaction of ANT and VDAC is thought to be essential for the regulation of PTP formation, [205,206]. This structure was suggested to include VDAC1 in the OMM, ANT in the IMM, and cyclophilin D (CyD) in the matrix [123,207,208]. However, mitochondria lacking all three VDAC isoforms retained an unaltered ability to undergo permeability transition [209], and an ANT knock-out study showed that it is not essential for PTP activity [210]. Recently, however, it was proposed that dimers of the ATP synthase complex can form the PTP [206].

VDAC1 also interacts with glycerol kinase (GK), c-Raf kinase, and the glycolytic enzyme glyceraldehyde 3-phosphate dehydrogenase, (GAPDH) [4,30,41,42,211]. Mitochondrial creatine kinase (MtCK) interacts with octameric VDAC1 [46] and lowers the affinity of VDAC1 for HK and Bax [212].

VDAC-tubulin interaction was proposed to serve as a metabolic switch to increase or decrease mitochondrial metabolism, ATP generation, and cytosolic ATP/ADP ratios [213].

Finally, other energy metabolism proteins that interact with VDAC1 include the OMM protein CPT1a that catalyzes the primary step of fatty acid oxidation [88,105], TSPO, which is involved in the transport of cholesterol into mitochondria [214], and the gluconeogenesis/glycolysis enzyme aldolase [91].

### 7.2. VDAC1 Interacting Apoptosis-Related Proteins

The Bcl-2 family of proteins comprises both pro-apoptotic members (e.g. Bid, Bax, Bim, and Bak), and pro-survival agents (e.g., Bcl-2, Bcl-xL, and Mcl-1) [215,216]. These proteins were shown to interact with VDAC1 to inhibit apoptosis [27,33,34,35,217,218,219,220,221]. Interaction of Bcl-2 and Bcl-xL with VDAC1 in a reconstituted membrane bilayer reduced the channel conductance of native, but not mutated VDAC1, and protected cells expressing native, but not mutated VDAC1 from apoptosis [27,34]. Site-directed mutagenesis was used to identify the VDAC1 domains that interact with Bcl-2 and Bcl-xL to purvey the anti-apoptotic effect [27]. The results indicated that the N-terminal region of VDAC1 was important for this interaction [4,23,27,33,38], and also for association with HK [33,39]. Another member of the family, Mcl-1 has been shown to directly interact with VDAC to increase mitochondrial Ca^2+^ uptake and ROS generation [217].

As a result of their ability to regulate the activity of both VDAC1 and IP_3_R, it is likely that the anti-apoptotic ability of Bcl-2, Bcl-XL, and Mcl-1 is mediated by the control of Ca^2+^ fluxes across the ER and mitochondrial membranes [28,222,223,224]. In this respect, we demonstrated that the BH4 domain of Bcl-XL, but not that of Bcl-2, selectively targets VDAC1 and inhibits apoptosis by decreasing VDAC1-mediated Ca^2+^ uptake into the mitochondria [225].

It has been shown that activating transcription factor 2 (ATF2), associated with cell death or cellular stress states in several melanoma and tumor cell lines, involves Bim and VDAC1, where VDAC1 depletion significantly prevented ATF2-related apoptosis [225]. In addition, ATF2 prevented HK interaction with VDAC1, sensitizing cells to apoptosis.

The inhibition of As_2_O_3_-, ethanol-, endostatin-, and cisplatin-induced apoptosis by siRNA has been interpreted by a number of studies to suggest that VDAC1 interacts with pro-apoptotic proteins Bax and Bak to allow Cyto *c* release [37,150,151,226,227].

The HK-VDAC1 interaction prevents release of pro-apoptotic factors such Cyto *c*, and subsequent apoptosis. Thus, HK plays a role in tumor cell survival via inhibition of apoptosis [24]. 

The recently reported interaction between TSPO and VDAC1 [228] is thought to play a role in the activation of the mitochondrial apoptosis pathway through TSPO involvement in the generation of ROS [229,230,231]. The grouping of TSPO molecules around VDAC1 is thought to increase ROS generation in the proximity of VDAC, leading to apoptosis induction [231,232]. In addition, by interaction with VDAC1, TSPO inhibits mitochondrial autophagy and contributes to the efficiency of mitochondrial quality control machinery [5,8,9,233], regulating mitochondrial structure and function [233,234]. 

Yet another observation of the pro and anti-apoptotic effects associated with VDAC1 is the inhibition of apoptosis by the phosphorylation of serine 193 of VDAC1 by NIMA-related protein kinase 1 (Nek1) [235,236] while the pro-apoptotic protein BNIP3 was shown to interact with VDAC1 to induce mitochondrial release of endonuclease G [237].

### 7.3. Interacting Cytoskeletal Proteins 

The cytoskeleton is involved in the regulation of bioenergetic functions in the cell with VDAC1 considered as the main regulator by interacting with several cytoskeletal proteins. Direct Ca^2+^-dependent binding of gelsolin (Gsn) to the C terminal of VDAC1 inhibits the activity of the VDAC1 channel and reduces the release of Cyto *c* from liposomes [238,239,240,241]. Binding to another member of the gelsolin family (adseverin) has a similarly anti-apoptotic effect [241].

The tubulin βII isoform in oxidative skeletal muscles and brain synaptosomes has been shown to act as a regulator of mitochondrial function by modifying the permeability of the VDAC channel for adenine nucleotides [242]. Tubulin was shown to associate with VDAC1 [242,243,244] and, at nanomolar concentrations, α and β tubulin heterodimers induce voltage-sensitive reversible closure of VDAC, reconstituted into planar phospholipid membranes [242]. This is proposed to sustain the Warburg effect [245]. The interaction between βII-tubulin and VDAC is thought to form a super-complex termed the mitochondrial interactome, composed of the ATP synthasome (ATP synthase, the respiratory system, and inorganic phosphate transporter), tubulin, VDAC1, and MtCK [246]. This construct can regulate mitochondrial respiration in adult cardiomyocytes and other metabolically active cells. The VDAC1-tubulin interaction, therefore, represents a new pharmacological target for the development of novel anti-cancer agents [213].

Other cytoskeleton-associated proteins that bind VDAC1 include the microtubule-associated protein 2 (MAP2) [247] and the dynein light chain (Tctex-1/DYNLT1) that is responsible for microtubule-based motile processes [248]. The function of the VDAC1–DYNLT1 interaction is not yet clear.

Actin was also shown to interact directly with VDACs, and also with another ten membrane channel proteins [249].

### 7.4. VDAC1 Interacting Signaling Proteins 

Association of endothelial NO synthase (eNOS) with VDAC1 upregulated eNOS activity. This increase was dependent on [Ca^2+^i]- [250].

Superoxide dismutase 1 (SOD1) is a predominantly cytosolic protein, with a mutant SOD1, associated with amyotrophic lateral sclerosis (ALS), found in mitochondria-rich fractions of cells [251,252,253]. This mutant SOD1 was found to bind VDAC1 reconstituted in a lipid bilayer and to inhibit the conductance of the VDAC1 channel [254]. Mutant SOD1 could also interact with Bcl2 and alter the interaction between Bcl-2 and VDAC1, thus, reducing OMM permeability [255,256].

The mitochondrial anti-viral signaling protein MAVS, also known as IFN-beta promoter stimulator-1 (IPS-1), virus induced signaling adaptor (VISA), or caspase activation recruitment domain adaptor inducing I FN-β (Cardif) [257] localized in the OMM, interacts with VDAC1 and modulates its stability via the ubiquitin–proteasome pathway [258].

Several additional proteins were also proposed to interact with VDAC1. These include PBP74, also known as mtHSP70/GRP75/mortalin [248], and CRYAB (α-crystallin B) [259]. The pre-synaptic protein α-synuclein interacts with the mitochondria via VDAC1 [260,261,262]. The nAChRs were identified in the mitochondria and proposed to regulate Cyto c release via interaction with VDAC [263,264].

α-synuclein is a neuron-specific protein localized in the presynaptic nerve terminals and nucleus. Although its exact function is still unknown, it is well established that α-synuclein is involved in synaptic activity through regulation of vesicle docking, fusion, and neurotransmitter release [265,266]. A number of studies have shown that the pre-synaptic protein α-synuclein interacts with the mitochondria via VDAC1 [260,261] and can pass through the VDAC1 channel to target complexes of the mitochondrial respiratory chain in the inner mitochondrial membrane [267].

### 7.5. Viral Proteins Interact with VDAC1 to Modulate Expression Level and Function

A number of studies have reported that certain viruses induce VDAC overexpression, or that some of their proteins interact with VDAC. Furthermore, silencing VDAC expression dramatically reduces the expression of viral proteins as summarized in Table 2 and in the following:Influenza A Virus: the virus attacks the respiratory system including the nose, throat, and lungs. One of the virus proteins interacting with VDAC1 [268] is influenza A virus protein (PB1-F2). The protein is localized to both mitochondrial membranes, OMM and IMM, and is a 90-amino-acid protein expressed from the +1 open reading frame in the PB1 gene of influenza A viruses. PB1-F2 contributes to the pathogenesis of the virus by binding to the mitochondria, altering their morphology, and inducing apoptosis via direct interaction with VDAC1 [268].Hepatitis B Virus (HBV): one of five known human hepatitis viruses: hepatitis A, B, C, D, and E. This virus causes hepatitis B in which the virus attacks the liver and can cause acute and chronic liver disease such as cirrhosis and hepatocellular carcinoma. Hepatitis B viral protein (HBx) is a multifunctional protein encoded by the HBV genome that stimulates HBV replication. HBx binds to the mitochondria and co-localizes with VDAC-1, where it alters the mitochondrial transmembrane potential by creating together a hexamer that affects mitochondrial physiology [269].Hepatitis E Virus (HEV): HEV infections are often asymptomatic, although they can be severe and cause fulminant hepatitis and extra-hepatic manifestations including neurological and kidney injuries. Chronic HEV infections may also occur in immunocompromised patients causing inflammation of the liver. The open reading frame 3 (ORF3) protein is a small phosphoprotein of 113 or 114 aa proposed to act as an adaptor to link the intracellular transduction pathways, reduce the host inflammatory response, and protect virus-infected cells [270]. ORF3 protein interacts with several of the host’s cellular proteins, including VDAC1, where in a cross-linking study, ORF3-expressing cells were shown to produce higher levels of oligomeric VDAC1 [174,271].Human Immunodeficiency Virus type 1 (HIV-1): the virus causes acquired immunodeficiency syndrome, with progressive breakdown of the immune system, and promotes life-threatening opportunistic infections. A viral protein R (Vpr) is a 96-amino-acid (14-kDa) protein. This protein stimulates virus transcription and interacts with the host’s proteins, playing an active role in viral pathogenic factors. Vpr protein induces cell cycle arrest in proliferating cells, as well as regulating activation and apoptosis of infected T-lymphocytes via interaction with VDAC1 [239].Dengue Virus (DENV): the virus causes dengue fever characterized by headache, severe muscle and joint pain, and upper respiratory symptoms. The protein, DENV E, is an envelope protein (E protein) which is the key component of the dengue virion. The ENV E protein plays a vital role in the viral lifecycle by mediating interaction with host cells and facilitating invasion. It has been reported that this protein interacts with various proteins in host cells, including with the cellular chaperone GRP78 along with mitochondrial VDAC1. Furthermore, down-regulation of VDAC by siRNA reduced the expression of several DENV proteins: NS1, NS3, NS5, and DENVE [272]. Thus, VDAC plays a significant role in DENV infection.The Infectious Bursal Disease Virus (IBDV): the virus causes immunosuppressive disease in young chickens. The VP5 protein, a non-structural protein of IBDV, is associated within the plasma membrane of the host-infected cells and plays an essential part in pathogenesis of IBDV. VP5 induces apoptosis in host cells via interaction with VDAC [273], with the VDAC inhibitor DIDS [53] decreasing VP5 expression and apoptosis [274]. On the other hand, silencing VDAC1 expression decreases the expression of VP1, VP2, and VP5 [275]. VP1 and VP2 are essential structural proteins that participate in IBDV capsid assembly and penetration into the host cell, as well as replication of virus. The VP2 carries neutralizing epitopes which control antibody-mediated neutralization of IBDV infection. VP2 acts as an apoptotic inducer in infected cells.Japanese Encephalitis Virus infection (JEV): the virus causes Japanese encephalitis. The JEV invades the CNS, resulting in neuroinflammation, which negates the neuroprotective role of microglia, as characterized by increased microglial activation and neuronal death. The JEV envelope protein (JEV-E) is the major structural protein that facilitates the viral infection by recognizing host cellular receptors and mediating membrane fusion. The primary target of this protein is to neutralize antibodies; therefore, it is a key virulence factor involved in pathogenesis. The JEV-E protein interacts with VDAC, and in response to JEV infection, VDAC1 is overexpressed and co-localized with the ER protein GRP78, a multifunctional chaperone protein (HSP70) increasing MAM [276].Ostreid herpes virus-1 (OsHV-1): OsHV-1 is a variant of herpes virus, and it has been a major threat to Pacific cupped oysters. It contributes to the pathogenesis of mass mortality disease in the early life-stage of oysters. Infected oysters exhibit an increase in glycolysis and increased VDAC1 expression, as revealed in both mRNA and protein levels, while this increase in VDAC1 levels correlates with susceptibility of oysters to OsHV-1 [277].

Thus, VDAC1-interacting protein complexes are formed under physiological and pathological conditions, mediating, and/or regulating metabolic, apoptotic, and other processes that may be impaired in disease.

## 8. VDAC1 Overexpression in Disease States and Association with Apoptosis 

Mitochondrial dysfunction has been implicated in many diseases including cancer, Alzheimer’s disease (AD), Parkinson’s disease (PD), amyotrophic lateral sclerosis (ALS), type 2 diabetes (T2D), and cardiovascular diseases (CVDs). VDAC1 is essential for proper mitochondrial function and, consequently, for normal cell physiology. Thus, an association of VDAC1 with various pathologies is only to be expected. In this context, VDAC1 has been shown to be over expressed in cancer [3,86], in the affected regions of AD brains [75,76,77], in β-cells in T2D [80,81,84], and in autoimmune diseases such as lupus [87], NASH [88], IBD (unpublished data), and in CVDs [89] (Figure 4). Since as already described, overexpression of VDAC1 induces apoptosis and cell death [29,51,99,146,147], this may represent a common mechanism in CVDs, AD, and T2D although it is currently a matter of debate whether the over expression is a cause or result of the pathology. 

Importantly, reducing the levels of VDAC1 with a specific siRNA, or our newly developed small molecules, e.g., VBIT-4 and VBIT-12, could inhibit VDAC1 oligomerization, correct the misdirection of the protein to the plasma membrane (PM) [84], and prevent mitochondria dysfunction and apoptosis [53] in both T2D [84] and models of lupus [87]. The following is a summary of VDAC1 overexpression in certain diseases.

### 8.1. Cancer and VDAC1 

Our understanding of cancer has recently seen a major paradigm shift towards the concept of cancer being a metabolic disorder. This notion was first introduced by Otto Warburg, who suggested that a hallmark of cancer cells is deregulation of cellular energy, and metabolism. Indeed, there are extensive reviews of the evidence supporting the general hypothesis that the main characteristics of cancer, including genomic instability and aerobic glycolysis, can be linked to impaired mitochondrial function and energy metabolism [281,282]. By regulating the metabolic and energetic functions of mitochondria, VDAC1 can, therefore, control the fate of cancer cells. VDAC1 is highly expressed in various tumors obtained from patients, and in tumors established in mouse models, as well as in cancer cell lines [3,15,85,86,283], providing supporting evidence for its significance in high energy-demanding cancer cells. Indeed, its pivotal role in regulating cancer cellular energy, metabolism, and viability is also underscored by the findings that abrogation of VDAC1 expression reduced cellular ATP levels, cell proliferation, and tumor growth [85,86,99,196]. The overexpressed VDAC1 contributes to cancer cell metabolism by facilitating the passage of essential metabolites, and delivering mitochondrial ATP directly to HK, which is also overexpressed in cancer (HK-I, HK-II) [5,284]. This fuels the high level of glycolytic flux seen in tumors, which have a high demand for metabolites or metabolite precursors. As a corollary, down-regulation of VDAC1 in tumor cells, with subsequent reduction in metabolite exchange between the mitochondria and cytosol, inhibited the growth of cells and tumors [43,85,86,99,196].

Interactions between VDAC1 and the anti-apoptotic proteins Bcl-2, Bcl-xL [27,33,34,285], and HK [33,39] protect tumor cells from apoptosis [33,39]. Reduction of VDAC1 expression by a specific miRNA, miR320a, promoted mitophagy in serum starved cervical cancer cells [286] and blocked tumor cell proliferation and invasion in NSCLC, both in vitro and in vivo [287]. Another miRNA species, miR-7, was shown to inhibit VDAC1 expression, proliferation and metastasis in hepatocellular carcinoma [288], possibly by affecting the PTP [289].

These results all suggest that VDAC1 could be a useful druggable target for anti-cancer therapy.

### 8.2. Over-Expression of VDAC1 in Neurodegenerative Diseases 

There is now emerging evidence connecting mitochondrial dysfunction to neurodegenerative disorders [290] with caspase-mediated apoptosis implicated in the premature neuronal cell death seen in neurological disorders (Table 3) [291,292,293,294,295,296].

A number of studies have linked VDAC dysfunction to AD [63,297,298,299], Down’s syndrome [299], and familial amyotrophic lateral sclerosis (ALS) [254,300]. High levels of VDAC1 were demonstrated in the dystrophic neurites of Aβ deposits in AD post-mortem patient brains and in brains of amyloid precursor protein (APP) transgenic mice [75,76,77]. Moreover, increases in VDAC levels in the thalamus of mice were shown to be associated with neurodegeneration in the Batten disease model [301]. Similarly spatial cognitive deficits in an animal model of Wernicke–Korsakoff syndrome were reported to be associated with changes in thalamic VDAC levels [302]. The involvement of plasmalemmal VDAC in AD was also proposed [62,63].

AD brains exhibit a significant loss of neurons, mainly due to apoptosis, with neurons in AD brains [31,64,303,304] displaying the hallmarks of apoptosis [305]. As VDAC1 overexpression was shown to trigger apoptotic cell death [29,51,99,146,147,148], we propose that overexpressed VDAC1 in the AD brain may be responsible for the observed neuronal cell death.

Proteomics studies have identified nitration and carbonylation of VDAC1 in the brains of AD patients, suggesting that VDAC1 channel activity contributes to the pathogenesis and progression of AD [317]. In addition, VDAC1-deficient transgenic mice exhibit deficits in long-term potentiation and learning behavior, suggesting that VDAC1 is an absolute requirement for normal brain activity [318].

Mutations in superoxide dismutase (SOD1) are the second most common cause of familial ALS, a progressive neurodegenerative disease characterized by the loss of motor neurons in the brain and spinal cord [319]. SOD1 mutants interfere with various aspects of mitochondrial function including mitochondrial energy metabolism, transport, fission, and fusion [320,321,322,323]. An inverse correlation between mutant SOD1 mitochondrial association in motor neuron-like NSC-34 cells and disease duration was reported in patients carrying mutations in SOD1 [324]. In addition, the misfolded SOD1 could also directly bind VDAC1, resulting in destabilization of VDAC1 conductance and channel instability, and inhibiting VDAC1 transport of adenine nucleotides across the OMM [254,306].

Recently [255], we demonstrated that mutant SOD1^G93A^ and SOD1^G85R^, but not wild-type SOD1, directly interact with VDAC1 and reduce its channel conductance, but no such interaction was obtained with N-terminal-truncated VDAC1. Moreover, a VDAC1-derived N-terminal peptide inhibited mutant SOD1-induced toxicity, suggesting that its specific interaction with the N-terminal domain of the VDAC1 is associated with mutant SOD1 toxicity.

In addition, proteins including SOD1, α-synuclein, and apoE, which all bind VDAC1 have been implicated in affecting intraneuronal Ca^2+^ in several neurodegenerative diseases, [7]. These findings suggest that VDAC1 could represent a potential target for novel therapeutic strategies also for neurodegenerative diseases.

### 8.3. Type 2 Diabetes (T2D) and VDAC1

VDAC is also overexpressed in T2D, which is the most common metabolic disease [325]. Elevated levels of VDAC were found in mouse coronary vascular endothelial cells isolated from diabetic mice [307] and in the β cells of T2D patients [80]. Interestingly, hyperglycemia increases VDAC1 expression in both pancreatic β-cells [80] and in the kidney [81]. Recently [84], we demonstrated up-regulation of VDAC1 in islets from T2D donors and in NS1 cells under glucotoxic conditions, with the result that VDAC1 is mislocalized to the plasma membrane of the insulin-secreting β cells, with loss of ATP. Specific anti-VDAC1 antibodies, and our VDAC1 inhibitor, VBIT-4, could restore generation of cellular ATP and normalize glucose-stimulated insulin secretion in T2D islets [84]. Similarly, treatment of db/db mice with VBIT-4 prevented hyperglycemia, and maintained normal glucose tolerance and physiological regulation of insulin secretion [84]. Thus, β-cell function was preserved by targeting the overexpressed VDAC1. Interestingly, lncRNA-H19/miR-675 was reported to regulate high glucose-induced apoptosis by targeting VDAC1, thus, providing a novel therapeutic strategy for the treatment of diabetic cardiomyopathy [326]. These findings point to the connection between VDAC, mitochondrial function, and the pathogenesis of T2D. The involvement of VDAC1 in T2D was also demonstrated in mice subjected to combination of a high-fat diet (HFD) and low-dose of streptozotocin (STZ), representing a model for type 2. These STZ/HFD-32 treated mice showed symptoms of T2D-like disease reflected in increased blood glucose [82].

### 8.4. Autoimmune and Inflammatory Diseases and Increased VDAC1 Expression Levels

Autoimmune diseases often are characterized by T-cell hyperactivity and β-cell overstimulation leading to overproduction of autoantibodies [327]. Inflammation is an important host defense response to injury, tissue ischemia, auto immune responses, and infectious agents [328]. This is now viewed as one of the major causes for the development of diseases, such as cancer, cardiovascular disease, diabetes, obesity, osteoporosis, rheumatoid arthritis, inflammatory bowel disease, asthma, and CNS- related diseases, such as depression and Parkinson’s disease.

Recently [87], we demonstrated that VDAC1, but not VDAC2, are over-expressed in diseases (such as SLE) that are associated with type-1 interferon signaling. SLE is a non-organ-specific autoimmune disease characterized by β cell hyperactivity, abnormally activated T cells, defects in the clearance of apoptotic cells, and immune complexes. We found that VDAC1 oligomers in the OMM promoted mtDNA release and our small molecule, VBIT-4, which inhibits VDAC1 oligomerization, decreased mtDNA release, type-Ι interferon signaling, neutrophil extracellular traps, and disease severity in a mouse model of systemic lupus erythematosus.

Additionally, we found that VDAC1 expression is upregulated in several autoimmune diseases, including Hashimoto thyroiditis, also known as chronic lymphocytic thyroiditis, an autoimmune disease in which the thyroid gland is gradually destroyed; rheumatoid arthritis (synovial), a persistent auto-immune disease primarily characterized by cytokine-mediated inflammation of the synovial inside layer of the joints and obliteration of cartilage and bone; psoriasis, an immune-mediated skin disease characterized by abnormal keratinocyte differentiation and proliferation; and Crohn’s disease, a chronic intestinal inflammatory condition caused by multiple factors. In addition to the non-immune contributing factors, the breach of the intestinal epithelial barrier and dysfunction of both innate and adaptive immunity that predominates during the inflammatory process is considered to be one of the earliest etiological factors involved in IBD. Still more diseases with VDAC1 overexpression are sarcoidosis, a disease involving abnormal collections of inflammatory cells that form lumps, with the disease usually beginning in the lungs, skin, or lymph nodes; and chronic granulomatous disease (CGD), an inherited primary immunodeficiency disease presenting with accumulation of immune cells at sites of infection or inflammation.

Representative images from healthy and diseased tissues and quantitative analysis of VDAC1 expression levels in the various pathological tissues are presented in Figure 5. These findings suggest that VDAC1 is a potential therapeutic target for lupus and other autoimmune diseases.

### 8.5. Cardiovascular Diseases, Apoptosis, and VDAC 

The involvement of VDAC1 in the pathogenesis of cardiac abnormalities has been proposed. In the context of cardiac I/R, upregulation of VDAC1 expression and phosphorylation have been shown to augment cardiomyocyte damage, and inhibition of these processes was mechanistically linked to the nutritional preconditioning function of resveratrol [309,311,312,313]. Oxidative stress damage in H9c2 myoblasts was reported to increase VDAC1 expression levels and its oligomerization [308,314]. Recently, it has been demonstrated that silencing VDAC1 promotes tBHP-induced apoptosis in H9C2 cells via decreasing mitochondrial HK-II binding and enhancing glycolytic stress [329]. In addition, VDAC1 was found to be involved in detrimental Ca^2+^ transfer from the ER to the mitochondria [106]. Upregulated transcriptional levels of a diverse array of genes including VDAC1 were found in the septal tissue of human patients with hypertrophic cardiomyopathy [310]. In addition, prominent upregulation of VDAC1 expressional levels in a rat model of cardiac hypertrophy induced by renal artery ligation and treatment with siRNA against VDAC1 partially inhibited the observed apoptotic cell death [259]. It was also reported that the expressional levels of VDAC1 were down-regulated in a cellular model of cardiomyocyte hypertrophy induced by the α1-adrenergic agonist phenylephrine that was prevented by peroxisome proliferator-activated receptor α (PPAR α) [330]. Recently, we demonstrated an increase in the expressed levels of VDAC1 in the setting of common cardiac pathologies including in the post-MI setup, chronic left ventricular (LV) dilatation and dysfunction, and hyperaldosteronism [90].

It is still unclear how the expression of VDAC1 is affected by common cardiac pathologies including pathological hypertrophy and ischemic cardiomyopathy. It is possible, however, that increased cardiomyocyte susceptibility to mitochondrial-mediated cell death is related to an increase in VDAC1 levels.

### 8.6. Non-Alcoholic Fatty Liver Disease (NAFLD) 

Chronic liver disease represents a significant public health problem world-wide, with viral hepatitis and non-alcoholic fatty liver disease (NAFLD) affecting about 20% of the general population [331]. NAFLD is characterized by excessive abnormal accumulation of fatty acids and triglycerides within the hepatocytes of non-alcohol users. NAFLD and the progressive state, non-alcoholic steatohepatitis (NASH) manifest as liver pathologies characterized by severe metabolic alterations due to fat accumulation that leads to liver damage, inflammation, and fibrosis [331]. Mitochondria play a prominent role in hepatosteatosis disease pathogenesis [332]. The involvement of VDAC in steatosis and NASH is suggested by several studies [83,88,333]. Mitochondria lacking VDAC1 do not oxidize fatty acids, and a VDAC1 inhibitor inhibited the oxidation of palmitate [334]. In addition, closure of the VDAC channel produces steatosis both in alcoholic steatohepatitis and NASH [83,333] because VDAC1 participates in the complex responsible for transporting fatty acids across the OMM [105,107].

Recently, we demonstrated that a cell-penetrating VDAC1-based peptide, R-Tf-D-LP4, arrested steatosis and the NASH produced by feeding mice a high-fat diet (HFD-32), and reversed liver pathology to a normal-like state [88]. R-Tf-D-LP4 treatment eliminated inflammation, liver fibrosis, and restored normal levels of glucose and liver enzymes. Peptide treatment also affected carbohydrate and lipid metabolism, increasing the expression of enzymes and factors associated with fatty acid transport to the mitochondria, enhancing β-oxidation and thermogenic processes, but decreasing the expression of enzymes and regulators of fatty acid synthesis [88] as well as HCC [283]. Thus, we suggest that R-Tf-D-LP4 peptide inhibits steatosis and NASH by increasing fatty acid oxidation via alterations in the liver transcriptional program; thus, it offers a promising therapeutic approach for steatosis and NASH.

### 8.7. Rheumatoid Arthritis, Acute Kidney and Spinal Cord Injury, and VDAC1 Expression

Rheumatoid arthritis (RA) is a chronic inflammatory disorder characterized by destructive polyarthritis, which destroys the joint synovial membrane, cartilage, and bone, resulting in disability [335]. Systemic inflammation mediated by cytokines is central to the pathogenesis of RA [336]. In a monkey model of RA [315], the heart receptor-interacting protein kinase 1 (RIPK1), a key pro-apoptotic signaling molecule, was shown to be upregulated, to bind VDAC1, and to promote oligomerization and subsequent cardiac cell death and functional impairment.

Acute kidney injury (AKI) is a clinical syndrome characterized by a rapid decline in kidney function and failure to regulate fluids, electrolytes, and the acid–base balance [337]. Mitochondrial dysfunction and ATP deficits are the most pronounced in tubular cells of the renal cortex and precede the clinical manifestations of AKI [338,339]. Ischemic insults could damage kidney function which then recovered in wild-type, but not in VDAC1-deficient mice. This suggests that VDAC1 regulates the recovery of renal mitochondrial function and dynamics and ATP levels, and increases survival after AKI [340]. VDAC oligomerization with subsequent release of Cyto *c* to the cytoplasm has also been implicated in cisplatin-induced apoptosis and nephrotoxicity [341].

In spinal cord injury (SCI), damage to the spinal cord can cause temporary or permanent changes in its function. Symptoms may include loss of muscle function, sensation, or autonomic function in the parts of the body served by the spinal cord. The primary injury immediately after trauma, may cause neuronal/oligodendrocyte cell death and axonal shear that is then followed by a secondary injury characterized by progressive pathology, including microvascular perfusion changes, inflammation, free radical generation, and apoptosis/necrosis [342]. This secondary injury often alters the metabolism of intact axonal tracts, as a result of extensive oligodendrocyte cell death. VDAC1 levels were shown to be significantly increased after spinal cord injury. Inhibition of VDAC1 oligomerization by DIDS, protects the spinal cord from demyelination and promotes locomotor function recovery after spinal cord injury [316]. Prevention of VDAC1 oligomerization might therefore be beneficial for clinical treatment of SCI.

In conclusion, considering that VDAC1 overexpression is associated with a variety of pathological conditions including cancer [3,86], AD [75,76,77], T2D [80,81], and the autoimmune disease lupus [87], understanding the regulatory mechanisms of VDAC1 overexpression and mislocalization to the PM may be clinically relevant to these diseases.

## 9. VDAC1-Based Strategies to Treat Diseases Such as Cancer, Neurodegenerative, Autoimmune, NASH, and T2D

Targeting VDAC1 is likely to be effective for conditions associated with altered cell metabolism and/or apoptosis and by VDAC1 overexpression [29,51,99,146,147], which we suggest may be a common mechanism in the pathology of CVDs, AD, and T2D. Alternatively, modulating VDAC1 to activate apoptosis could be a possible therapeutic strategy for cancer. Thus, a new generation of VDAC1-based therapeutics may impact the treatment of a wide variety of diseases.

VDAC1-silencing strategy—we recently demonstrated that using specific si-RNA to silence human VDAC1 (si-hVDAC1) in cultured cells and in mouse models reduced cell energy homeostasis and inhibited the growth of various cancer cell types including glioblastoma multiforme (GBM), lung cancer, and triple negative breast cancer [85,86,343]. “Tumor” cells remaining after the treatment appeared to have undergone re-programmed metabolism and cell differentiation into normal-like cells with inhibited angiogenesis, epithelial mesenchymal transition (EMT), invasiveness, and stemness. Depletion of VDAC1 altered the expression of hundreds of genes including transcription factors (TFs) that regulate signaling pathways associated with cancer [85]. Interestingly, this re-programming was related to the time during which the tumor cell was depleted of hVDAC1, suggesting that a chain of events is involved [344].

Silencing VDAC1 expression by specific siRNA in PC12 cells and SH-SY5Y cells prevented Aβ entry into the cytosol, and reduced Aβ-induced toxicity [31]. These results suggest the involvement of VDAC1 in Aβ-cell toxicity, raising the possibility that Aβ enters the cell via plVDAC1 [31].

VDAC1-based peptides–A hallmark of cancer cells is their ability to avoid apoptosis [345,346] by overexpressing anti-apoptotic proteins such as HK-I, HK-II, Bcl2, and Bcl-xL [201,284,347,348,349,350,351,352], which interact with VDAC1 to prevent apoptosis [4,23,24,26,27,29,33,34,35,36,37,38,39,201,202,204,353]. We have designed a number of cell-penetrating VDAC1 peptides that target amino acids important for the interactions with HK, Bcl-2 and Bcl-xL as identified by point mutations and consideration of the VDAC1 domains [24,27,29,33,34,39,199]. Two of these VDAC1-based peptides, Tf-D-LP4 and the VDAC1-N-terminus, could induce cell death in a variety of cancer cell lines, regardless of cancer type or mutation status, but with a definite specificity for cancerous cells [33,39,199].

Tf-D-LP4 in treating cancer—The peptides were used in several disease mouse models. Tf-D-LP4 crossed the blood-brain barrier in a GBM mouse model to inhibit tumor growth by decreasing the energy metabolism while inducing apoptosis and also upregulating pro apoptotic proteins [200]. Similar effects were seen in lung and breast cancer [354]. Thus, the peptide has the potential to serve as the basis for new anti-cancer therapies.

Tf-D-LP4 in treating steatosis and NASH—the peptide was also found to be effective in treating non-alcoholic fatty liver disease (NAFLD), as induced by treating mice with streptozotocin (STZ) and a high-fat diet (STZ/HFD-32). R-Tf-D-LP4 treatment eliminated inflammation, liver fibrosis, and normalized the liver enzymes [88]. In addition, the peptide increased the expression of proteins associated with fatty acid transport to the mitochondria, and enhanced β-oxidation and thermogenic processes, while decreasing the expression of enzymes and regulators of fatty acid synthesis [88]. The VDAC1-based peptide, thus, offers a promising therapeutic approach for steatosis and NASH.

The bold letters indicate the cell-penetrating peptide sequence (Antp, Tf), the italic letters indicate amino acids involved in the tryptophan zipper (hairpin formation), and the underlined sequences represent amino acids in D conformation.

Tf-D-LP4 in treating type 2 diabetes (T2D)–STZ/HFD-32 mice also displayed symptoms of a T2D-like disease. R-Tf-D-LP4 peptide treatment restored the elevated blood glucose to close to normal levels, and increased the number and average size of islets and their insulin content, as compared to untreated controls [82]. Moreover, peptide treatment of STZ/HFD-32 fed mice increased the expression of β-cell maturation and differentiation, PDX1 transcription factor, which enhanced the expression of the insulin-encoding gene, and is essential for islet development, function, proliferation, and maintenance of glucose homeostasis in the pancreas. These results suggest that the VDAC1-based R-Tf-D-LP4 peptide has potential as a treatment for diabetes.

VDAC1-N-terminal-Antp peptide in treating cancer–The VDAC1 N-terminal domain (26 amino acids) is required for interaction with VDAC1-associated proteins and apoptosis [33,39]. A number of VDAC1 N-terminal peptides fused to Antp (Penetrating), a 16-residue-long sequence from the Drosophila antennapedia homeodomain were prepared (Table 4). One of these, the D-Δ(1-14)N-Ter-Antp peptide, composed of the VDAC1 N-terminal region (15–26 amino acids), and Antp was shown to induce apoptosis in a variety of cancer cell lines [354].

VDAC1-N-terminal peptide (N-Ter)—the peptide without the cell penetrating sequence ANTP directly interacted with Aβ, and prevented Aβ cellular entry and Aβ-induced mitochondria-mediated apoptosis [31].

VDAC1 N-terminal peptides prevent mutant SOD1 toxicity–A VDAC1-derived N-terminal peptide version that did not induce cell death, inhibited mutant SOD1-induced toxicity. The peptide enhanced survival of motor neuron-like NSC-34 cells expressing mutant SOD1 [255]. VDAC1 N-terminal peptides targeting mutant SOD1 may represent potential new therapeutic strategies for ALS.

VDAC1-interacting small molecules, VBIT-4 and VBIT-12, treating diseases with overexpressed VDAC1—our newly developed small molecules, VBIT-4 and VBIT-12, were used to prevent mitochondria dysfunction and apoptosis in conditions thought to be caused by VDAC1 over expression and oligomerization [53]. The effect of VBIT-4 was validated in T2D [84] in lupus models [87] and on fibrosis induced by hyperaldosteronism in the heart [90], and the efficacy of VBIT-12 in DSS-induced colitis (unpublished data).

VDAC inhibitor DIDS inhibits the effects of IBDV [53]—as described in Section 7.5, DIDS decreased the expression of VP5 and apoptosis [274] induced by IBDV, causing immunosuppressive disease in young chickens and up-regulated VDAC1 expression in cells.

In addition, although not a specific inhibitor of VDAC1 oligomerization, DIDS could protect rats from the secondary response to spinal cord injury [54]. In this model, DIDS reduced oligodendrocyte cell death, and improved axonal density, to promote motor function recovery.

## 10. Conclusions 

Mitochondria represent the energy hub of the cell, and their dysfunction plays a critical role in tumorigenesis and a broad range of other pathologies including AD, CVDs, T2D, and a wide variety of autoimmune diseases, which are characterized by over-expression of the VDAC1 protein. The ability of VDAC1 to control transport through the mitochondrial membrane gives the protein the ideal opportunity to serve as a gatekeeper, orchestrate the traffic of metabolites, and regulate apoptosis, and other cell stress-associated processes. The strategic location in the OMM also makes VDAC1 well positioned to interact with over 100 proteins, allowing it to mediate and regulate the integration of mitochondrial and cellular functions, significantly by modulating Ca^2+^ homeostasis. One of the most important protein interactions is with itself since the dynamic oligomerization of VDAC1 is seen to be a pivotal regulation point in apoptosis regulation and the decision of cell fate. These, together with its overexpression in cancer and other diseases, including Alzheimer’s disease, some cardiovascular diseases and type 2 diabetes, suggest that VDAC1 overexpression is associated with cell response to stress conditions. As treatments that affect the degree of expression and/or oligomer formation of VDAC1 have been efficacious in a wide variety of models, targeting VDAC1 has vast therapeutic potential to modulate the biology of cancer and other diseases.

## Figures and Tables

**Figure 1 biomolecules-10-01485-f001:**
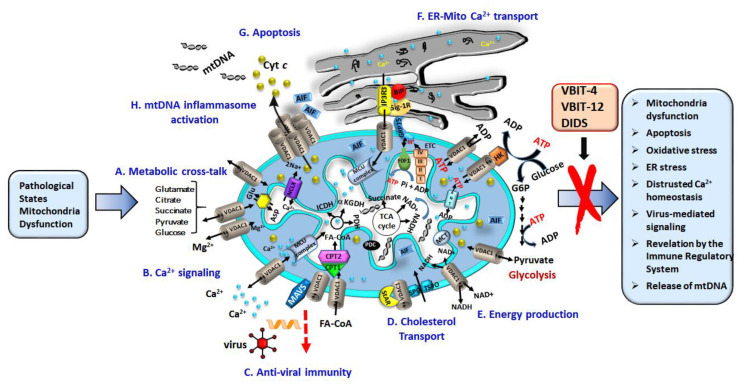
Voltage-dependent anion channel 1 (VDAC1) as a multi-functional channel mediates metabolites, nucleotides, and Ca^2+^ transport, controlling energy production, endoplasmic reticulum (ER)-mitochondria cross-talk, and apoptosis. VDAC1 is responsible for a number of functions in the cell and mitochondria including: (**A**) transfer of metabolites between the mitochondria and cytosol; (**B**) passage of Ca^2+^ to and from the intermembrane space (IMS) to facilitate Ca^2+^ signaling; (**C**) Mitochondrial antiviral-signaling protein (MAVS) associated with VDAC1 enable anti-viral signaling. (**D**) Transfer of acetyl coenzyme-A (acyl-CoAs) across the outer mitochondrial membrane (OMM) to the IMS, for conversion into acylcarnitine by CPT1a for further processing by β-oxidation. Together with Star and translocator protein (TSPO), VDAC1 forms the multi-protein transduceosome, which transports cholesterol. (**E**) Recycling ATP/ADP, NAD+/NADH, and acyl-CoA between the cytosol and the IMS, and regulating glycolysis via association with HK; (**F**) contributing to ER-mitochondria contacts, where Ca^2+^ released by IP3 activation of inositol 1,4,5-trisphosphate receptors (IP_3_R) in the ER is directly transferred to IMS via VDAC1, and then is transported to the matrix by the Ca^2+^ uniporter (MCU complex). In the matrix Ca^2+^ regulates energy production via activation of the tricarboxylic acid cycle (TCA) cycle enzymes: pyruvate dehydrogenase (PDH), isocitrate dehydrogenase (ICDH), and α-ketoglutarate dehydrogenase (α-KGDH). The electron transport chain (ETC) and the ATP synthase (FoF1) are also presented. (**G**) VDAC1 oligomers forming a hydrophilic protein-conducting channel capable of mediating the release of apoptogenic proteins (e.g., Cyto c and apoptosis-inducing factor (AIF)) from the mitochondrial IMS to the cytosol, leading to apoptosis. (**H**). VDAC1 oligomers allow mtDNA release triggering inflammasome activation. Pathological conations lead to dysfunction of the mitochondria as reflected in the activities presented in the box on the right. These altered activities can be prevented by VDAC1-interacting molecules, such as DIDS, VBIT-4 and VBIT-12.

**Figure 2 biomolecules-10-01485-f002:**
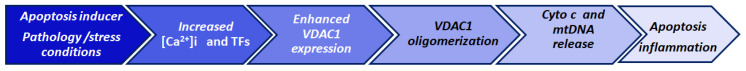
Sequence of events leading to VDAC1 overexpression, oligomerization, and apoptosis. Apoptosis stimuli or pathological conditions enhance VDAC1 expression via Ca^2+^ or transcription factors (TFs) activating promoter, leading to VDAC1 transcription.

**Figure 3 biomolecules-10-01485-f003:**
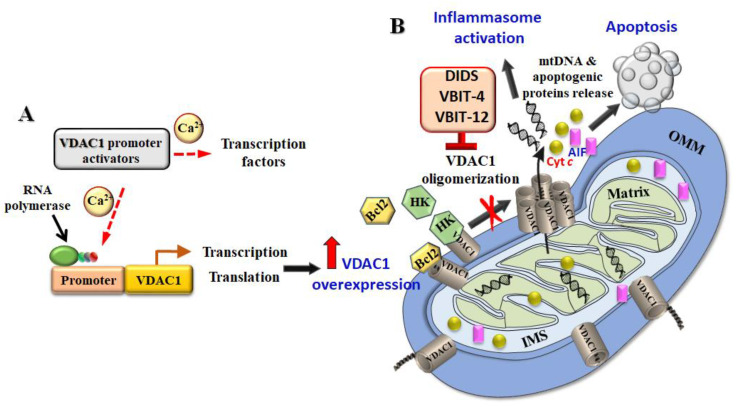
Proposed model for the mechanism of action for activators of the VDAC1 promoter leading oligomerization, and apoptosis. (**A**) Apoptotic stimuli act either by directly activating the VDAC1 promoter or by stimulating the activity of TFs, thereby inducing VDAC1. Ca^2+^ directly or via a TF leads to promoter activation. (**B**) Overexpressed VDAC1 shifts the equilibrium to the VDAC1 oligomeric state, mediating the release of apoptogenic proteins, and leading to apoptosis. VDAC1-interacting molecules such as VBIT-4, inhibit VDAC1 oligomerization and apoptosis.

**Figure 4 biomolecules-10-01485-f004:**
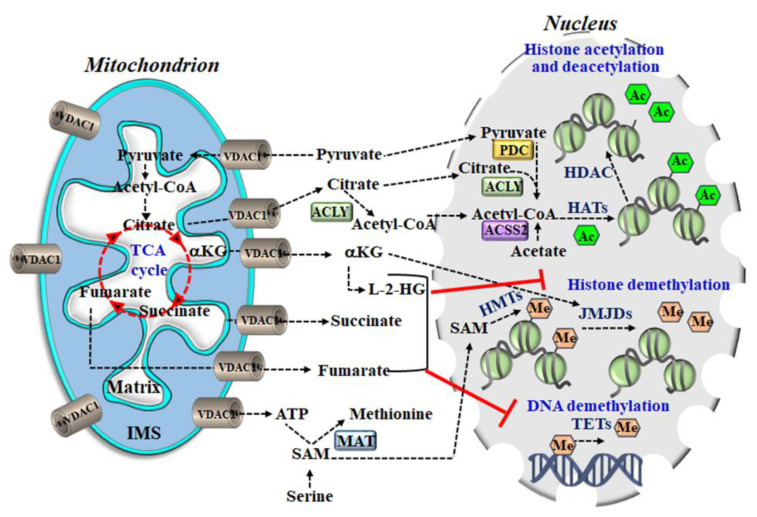
Mitochondrial metabolites are transported by VDAC1 to the cytosol and are then transported to the nucleus where they serve as substrates for enzymes that modify chromatin. Mitochondrial metabolic pathways including the TCA cycle generate substrates required for the methylation, acetylation, or demethylation reactions that modify chromatin. Specifically, histone acetylation by histone acetyltransferases (HATs) is dependent on the availability of acetyl groups provided by acetyl-CoA, which is produced in the cytosol by ACLY using citrate exported from the TCA cycle in mitochondria. α-ketoglutarate (α-KG) is an essential co-factor of 2-oxoglutarate-dependent dioxygenases (2-OGDD), including the histone demethylases Jumonji domains (JMJDs) and ten-eleven translocations (TETs), which are DNA demethylases. Succinate is the product of 2-OGDD enzyme reaction; thus, when it accumulates, it works as an antagonist of the reaction. Moreover, 2-Hydroxyglutarate (2-HG) and fumarate can also rewire the epigenetic landscape of the cells through inhibition of histone and DNA demethylation. The metabolites cross the IMM via specific transporters, and cross the OMM via a single protein VDAC1. The metabolites directly relevant to chromatin regulation reach the nucleus where several metabolic enzymes are localized including: methionine adenosyl-transferase (MAT); ATP-citrate lyase (ACLY), which catalyzes the ATP-dependent cleavage of mitochondrial-derived citrate into oxaloacetate; acetyl-CoA; pyruvate dehydrogenase complex (PDC); and acetyl-CoA synthetase 2 (ACSS2).

**Figure 5 biomolecules-10-01485-f005:**
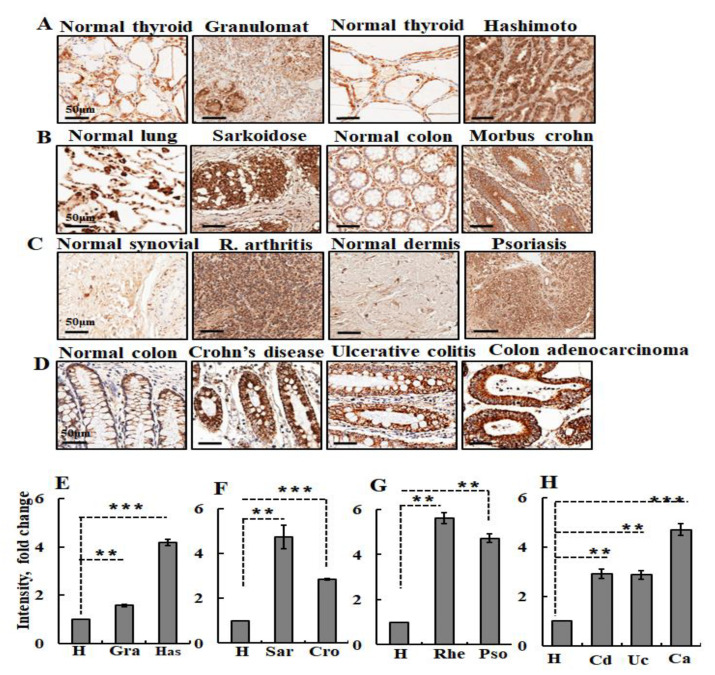
VDAC1 is overexpressed in tissue samples from autoimmune disease patients. (**A**–**D**) immunohistochemistry staining (IHC) of VDAC1 was performed on tissue microarray slides obtained from Provitro AG (Charité Campus Mitte 10,117 Berlin, Germany) (**A**–**C**) and US Biomax, Inc. (MD USA) (**D**). The array contains tissue sections from healthy (normal thyroid, lung, colon, synovial, dermis samples), (**A**) granulomat (Gra), Hashimoto thyroiditis (Thyroid, Has), (**B**) sarcoidosis (lung, Sar), Morbus Crohn (Sigma), (**C**) rheumatoid arthritis (Synovial, Rhe), psoriasis (Pso), (**D**) Crohn’s disease (Cd), ulcerative colitis (Uc), colon adenocarcinoma (Ca). (**E**–**H**) Quantitative analysis of VDAC1 expression levels are also presented. Representative sections of the indicated tissues were incubated overnight at 4 °C with anti-VDAC1 antibodies diluted in 1% BSA in phosphate buffered saline (PBS) and then with secondary antibodies diluted in 1% BSA in PBS. The slides were subsequently treated with 3’3-diaminobenzidine tetra-hydrochloride (DAB) and counter-stained with hematoxylin. Negative controls were incubated without primary antibody. Sections of tissue were observed under an Olympus microscope, and images were taken at 200× magnification with the same light intensity and exposure time. Quantitation of VDAC1 expression levels in the whole area of the provided sections, as reflected in the staining intensity, was performed using a panoramic microscope and HistoQuant software (Quant Center 2.0 software, 3DHISTECH Ltd). The results are the mean values ± SEM, ** *p* < 0.01, *** *p* < 0.001.

**Table 1 biomolecules-10-01485-t001:** Compounds and conditions inducing VDAC1 overexpression and cell death.

No.	Compound	Ref.	No.	Compound	Ref.
1	Prednisolone1	[168]	11	Etoposide	[51]
2	Cisplatin	[51,169]	12	Selenite	[51]
3	Arsenic trioxide (As_2_O_3_)	[51]	13	A23187	[51]
4	Arbutin (hydroquinone -O-β-D glucopyranoside)	[167,172]	14	Somatostatin	[65,173]
5	Hepatitis E virus ORF3 protein	[174]	15	H_2_O_2_	[51]
6	UV irradiation	[175]	16	Thapsigargin	[51]
7	Mechlorethamine and its derivative, melphalan	[176]	17	Ionomycin	[51]
8	Vacuolating cytotoxin	[177]	18	Vorinostat	[51,178]
9	Quinocetone	[179]	19	DSS	(unpublished data)
10	Endostatin	[151]	20	Pyrroloquinoline quinone (PQQ)	[171]
			21	Plasminogen kringle 5 (PK5)	[74]

**Table 2 biomolecules-10-01485-t002:** Different viruses target VDAC to modulate mitochondrial activities-VDAC1 is a key protein in the viruses’ effects on the host cells.

Virus	Virus Type	Viral Protein	Action on VDAC1	Reference
HIV-1 Human Immunodeficiency Virus type 1	Single-stranded positive RNA	Vpr	Interacts with VDAC1 and induces apoptosis	[239,278]
DENV Dengue Virus	Single-stranded positive RNA	DENV-E	Interacts with GRP78, a chaperone interacting with VDAC1. Down-regulation of VDAC reduced DENV protein expression	[272]
Influenza A Virus	Single-stranded negative-sense RNA	PB1-F2	Induces apoptosis via interaction with VDAC1	[268]
IBDV Infectious Bursal Disease Virus	Double-stranded RNA	VP5	Induces apoptosis in DF-1 cells via interaction with VDAC1	[53,273,274,279]
HBV Hepatitis B Virus	DNA	HBx	HBx interaction with VDAC1 to form a hexamer, causing mitochondria dysfunction	[269,280]
HEV Hepatitis E Virus	Single-stranded positive sense RNA	ORF3	Expression of ORF3 increases VDAC1 levels, siRNA against ORF3 reduces VDAC1 levels	[174,271]
Japanese Encephalitis Virus	Single-stranded, positive-sense RNA	JEV E	JEV E protein interacts with VDAC. Upon JEV infection, VDAC1 was co-localized with the ER protein GRP78 (HSP-70)	[276]
OsHV-1 Ostreid Herpes Virus-1	DNA		Increases VDAC1 expression at the mRNA and protein levels	[277]

**Table 3 biomolecules-10-01485-t003:** VDAC1 association with diseases.

Disease	VDAC1 State	Function	Reference
Cancer	Overexpressed	Increased cancer cell metabolic activity	[3,15,85,86,99,196,283]
Alzheimer’s disease	Overexpressed	Neuronal cell death	[75,76,77]
ALS—Amyotrophic lateral sclerosis	Interacted with mutated SOD	Enhanced cell survival	[254,255,306]
Type 2 diabetes	Overexpressed	Impaired generation of cellular ATP	[80,81,84,307]
Systemic lupus erythematosus	Overexpressed	Mediated release of mtDNA and triggered type-Ι interferon responses	[87]
Cardiovascular diseases	Overexpressed	Cardiomyocyte cell death	[90,308,309,310,311,312,313,314]
Non-alcoholic fatty liver disease (NAFLD)	Overexpressed	Mediated transport of fatty acids across the OMM	[88,105,107]
Rheumatoid arthritis	Overexpressed	Cardiac cell death and functional impairment	[315], Figure 5
Acute kidney injury	Deletion of VDAC1 hinders recovery of mitochondrial and renal functions	Required for the recovery of renal mitochondrial function	[315]
Spinal cord injury	Overexpressed	Oligomerization and apoptosis	[316]

**Table 4 biomolecules-10-01485-t004:** Amino acid sequences and analytical data for the VDAC1-based peptides.

Peptide	Sequence	No. of AA	Molecular Mass, Da
Tf-D-LP4	HAIYPRH*SWTWE*-199-KKLETAVNLAWTA GNSN-216-*KWTWK*	34	4111
R-Tf-D-LP4	*KWTWK*-216-NSNGATWALNVATELKK-199-*EWTWS*HRPYIAH	34	4111
N-Ter	1-MAVPPTYADLGKSARDVFTKGYGFGL-26-	26	2762
N-Ter-Antp	1-MAVPPTYADLGKSARDVFTKGYGFGL-26-RQIKIWFQNRRMKWKK	42	4990
D-Δ(1-14)N-Ter-Antp	15-RDVFTKGYGFGL-26-RQIKIWFQNRRMKWKK	28	3588
ΔN-Ter Δ(21-26)-Antp	1-MAVPPTYADLGKSARDVFTK-20-RQIKIWFQNRRMKWKK	36	4396
Δ(1-4)N-Ter Δ(21-26)-Antp	5-PTYADLGKSARDVFTK-20-RQIKIWFQNRRMKWKK	32	3997
Δ(1-9)N-Ter Δ(21-26)-Antp	10-LGKSARDVFTK-20-RQIKIWFQNRRMKWKK	27	3450

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
