# Peer review of "VDAC1 at the Intersection of Cell Metabolism, Apoptosis, and Diseases"

_biomolecules, 2020, doi:10.3390/biom10111485_

Round 1
Reviewer 1 Report
The review of Shoshan-Barmatz "VDAC1 at the Intersection of Cell Metabolism, Apoptosis and Diseases" gives a broad overview of VDAC functions and research of the role of VDAC in several pathologies.
The review is well organized and structured. All necessary background information is given in the first parts of a review, afterwards there is a description of the VDAC action in the molecular and cellular levels (involvement in cell death, metabolism, bioenergetics, gene expression, intracellular signaling). Then the authors switch to the pathological conditions related to real diseases. Overall this is a great review. Besides, the overview of the existed research data authors provides some models of mechanisms of VDAC1 actions, which are illustrated well. All the conclusions related to the mechanisms of VDAC1 action in disease-related conditions also confirmed by a massive overview of literature data, tables, and schemes.
I have only some small suggestions to the authors. I noticed only some typos all over the manuscript, I would suggest to the authors doublecheck it. Also, it seems to me that it would be nice to provide some brief info about different types of VDAC in the very beginning, where the authors just introduce 3 of them. In lines 67-69 I met some not really clear sentence "Interestingly when detected in plasma membrane...". Please rephrase it, or add some info to it.
Overall the manuscript is great and worth publishing.
Author Response
We thank this reviewer for his/her comments. The manuscript has been checked for grammar, punctuation, and spelling.
We have addressed the comment concerning VDAC1 isoforms, adding:
Information about VDAC isoform function and structure was obtained from channel activity of purified and reconstituted protein and, using cell-based assays for survival, metabolism, ROS, and cellular Ca2+ regulation, and by gene knockout mouse models [17]. VDAC1 is the most abundant isoform and the focus of this review. VDAC2 KO is lethal and is considered to be an anti-apoptotic protein. While VDAC 1 contains 2 cysteines, VDAC2 and VDAC3 with 9 and 6 cysteines, respectively, are proposed to function as oxidative stress sensors [17] .
The unclear sentence is now rephrased
Reviewer 2 Report
Comments attached

Author Response
Suggestions:
1) It would be nice to highlight the controversy related to VDAC as an mPTP. Although it has been mentioned in the review that ANT, VDAC1 and Cyclophilin D form the PTP complex, further I would suggest to include the study wherein it was shown that genetic knock out of all the three VDAC isoforms did not alter the mPTP opening (Voltage-dependent anion channels are dispensable for mitochondrial-dependent cell death., Baines et al 2007).
The study by Baines et al. was added: However, mitochondria lacking all three VDAC isoforms retained an unaltered ability to undergo permeability transition [209], and an ANT knock-out study showed that it is not essential for PTP activity [210].
2) A recent article on how silencing VDAC1 promotes apoptosis in H9C2 cells via decreasing mitochondrial HKII binding and increasing glycolytic stress (PMID: 32901466) can be included under the section titled as “Cardiovascular diseases, apoptosis, and VDAC”.
We added: Recently, it has been demonstrated that silencing VDAC1 promotes tBHP-induced apoptosis in H9C2 cells via decreasing mitochondrial HK-II binding and enhancing glycolytic stress [329].
3) The review also mentions engineered VDAC peptides that has a potential to be used as a therapeutic agents. It would be nice to introduce a table highlighting the sequence of those peptides, the mechanism by which they modulate the different diseased condition such as cancer and diabetes.
As suggested, we added a Table (Table 4) presenting the peptides indicated in this review with respect to sequences and some properties.
Reviewer 3 Report
In their Manuscript “VDAC1 at the intersection of cell metabolism, apoptosis, and diseases” the authors present an extensive collection of knowledge on the various roles of VDAC1 in different cellular processes and the resulting effects in health and disease.
While this work attempts to summarise the vast knowledge on VDAC1, unfortunately the readability of the manuscript suffers greatly. In its present form, it is hard for the reader to take away significant information as there is a certain degree of information overload. A more limited review (or multiple ones) focussing on specific aspects of VDAC1 would be more approachable to the reader and give the possibility to detach the content from a slightly more elaborate enumeration of facts/publications to an actual discussion of specific aspects of VDAC1 biology and function.
The high degree of information in limited space is also evident in the Figures, which try to convey too much information in too little space. This could be mitigated by an increase in size, separation into multiple panels and a general effort in decluttering.
Finally, the manuscripts suffers from a number of little mistakes, both in terms of grammar, punctuation and spelling as well as in the references (there are two references cited as “in progress” and one submitted, the latter actually having been published in 2019. Given the large number of references mistakes are hard to avoid but should be corrected in a final version).
Author Response
While this work attempts to summarise the vast knowledge on VDAC1, unfortunately the readability of the manuscript suffers greatly. In its present form, it is hard for the reader to take away significant information as there is a certain degree of information overload. A more limited review (or multiple ones) focussing on specific aspects of VDAC1 would be more approachable to the reader and give the possibility to detach the content from a slightly more elaborate enumeration of facts/publications to an actual discussion of specific aspects of VDAC1 biology and function.
- It should be noted that this article is for the special issue on:
(a) In this issue, an article from Prof. De Pinto’ group is also included: “Renaissance of VDAC: new structural insights on a protein hub at the interface between mitochondria and cytosol”. Thus, we tried to prevent overlapping with it.
(b) In the last years (2017–2019), reviews on different aspects of VDAC1 structures [1], functions in cell stress [2], Ca2+ regulation and ER-mitochondria communication [3, 4], metabolism [2], apoptosis [3, 5], and as a therapeutic target [6-11] were published by us and others. This is now added to the review abstract.
(c ) Here, we focused on VDAC1 overexpression, as induced by apoptosis inducers and pathological conditions, and summarized all reports connecting VDAC1 to many diseases and to the effects of 8 reported viruses whose proteins either interact with VDAC1, modulate its expression levels, or show that VDAC1 depletion inhibited the expression of several viral proteins in the infected cells.
The high degree of information in limited space review is also evident in the Figures, which try to convey too much information in too little space. This could be mitigated by an increase in size, separation into multiple panels and a general effort in decluttering.
Most of the figures for review are designed to summarize and present a concept. For example, Fig. 1 presents the multi-functions of VDAC1 in regulating mitochondria activities and to where the mitochondria dysfunction leads. Fig. 3 represents the sequence of events associated with VDAC1 overexpression, and Fig. 4 presents the link between mitochondria-generated metabolites and epigenetic modifications.
We do not see how the figures can be separated into several panels. The size of the figures is large enough for viewing them.
Finally, the manuscript suffers from a number of little mistakes, both in terms of grammar, punctuation and spelling as well as in the references (there are two references cited as “in progress” and one submitted, the latter actually having been published in 2019. Given the large number of references mistakes are hard to avoid but should be corrected in a final version).
The MS was checked for grammar, punctuation, and spelling by an English expert.
As for the references, we are sorry about this. We should have checked carefully the endnote generated references when selecting the journal format. These are now corrected
Round 2
Reviewer 3 Report
Thanks to the authors for submitting a revised version of their manuscript “VDAC1 at the intersection of cell metabolism, apoptosis, and diseases”. The authors have made an admirable job in collecting massive amounts of literature on VDAC1 and trying to assemble it into a review.
I appreciate their effort to correct the small mistakes in the references. Unfortunately they failed to address my main point of criticism and made only a minimal effort in trying to prepare a more focussed review. In its present form, the manuscript (including the Figures) is very inaccessible to readers, especially those not familiar with the field. While I certainly appreciate the effort the authors have put into researching the relevant literature, the results from this literature review is simply overwhelming to the reader.